# Comparing Different Packaging Conditions on Quality Stability of High-Pressure Treated Serra da Estrela Cheeses during Cold Storage [note 1]

**DOI:** 10.3390/foods12101935

**Published:** 2023-05-09

**Authors:** Rita S. Inácio, Maria J. P. Monteiro, José A. Lopes-da-Silva, Ana M. P. Gomes, Jorge A. Saraiva

**Affiliations:** 1LAQV-REQUIMTE, Department of Chemistry, Campus Universitário de Santiago, University of Aveiro, 3810-193 Aveiro, Portugaljals@ua.pt (J.A.L.-d.-S.); 2CBQF—Centro de Biotecnologia e Química Fina—Laboratório Associado, Universidade Católica Portuguesa, Escola Superior de Biotecnologia, Rua Diogo Botelho 1327, 4169-005 Porto, Portugal

**Keywords:** ewe raw milk cheese, high pressure, vacuum, wrapped in paper, plastic film, microbial evolution, safety, storage, sensorial, proteolysis

## Abstract

*Serra da Estrela* cheese with a Protected Designation of Origin (PDO) is a traditional cheese that is wrapped in paper without vacuum. High-pressure processing (HPP), which requires vacuum packaging of the cheese, has been used for its cold pasteurization to overcome safety issues. In this study, two packaging systems were studied: non-vacuum greaseproof paper wrapping package and vacuum packaging in plastic film. *Lactococci*, *lactobacilli*, enterococci, and total mesophiles reached ca. 8 log cfu g^−1^ and 4–6 log cfu g^−1^ in control (unpasteurized) and HPP-treated cheeses, respectively, with no significant differences between packaging systems. Spoilage microorganisms' viable cell numbers were reduced to <3 log cfu g^−1^ (quantification limit) in HPP-treated cheeses, independently of the packaging system. Yeasts and molds reached >5 log cfu g^−1^ in non-vacuum paper-wrapped cheeses. A vacuum-packaging system enabled better control of cheese proteolysis, which was revealed to be closer to that of the original control cheese values at the end of the 10-month storage period. In addition, cheese stored under vacuum film packaging became harder than non-vacuum paper-wrapped cheeses at each time point. Overall, conventional non-vacuum paper wrapping is adequate for short storage periods (<3 months), but for long periods vacuum packaging in plastic film is preferable.

## 1. Introduction

Traditional cheeses made from raw milk have been manufactured for centuries and are well known for their unique organoleptic characteristics. A common example is the Portuguese *Serra da Estrela* Cheese with Protected Designation of Origin (PDO) certification in the European Union [1]. For the production of this cheese, only three ingredients are used: raw ewes’ milk, crude extract of *Cynara cardunculus* L. as a coagulant, and salt [1,2]. Since milk is not thermally pre-treated, both spoilage and pathogenic microorganisms may be found in this type of cheese if not properly handled [3], which raises safety issues. Current efforts are constantly directed at meeting the needs of consumers by ensuring cheese quality with prolonged shelf-life during storage and commercialization. Two possible ways are related to milk non-thermal pasteurization and protective packaging systems. High-pressure processing (HPP) has been studied and applied as an alternative non-thermal process to conventional thermal pasteurization, with a 2–3-fold shelf-life extension for dairy foods compared to the raw counterparts [4], and with low impact on quality. Several studies have revealed the advantage of HPP application at the optimum organoleptic ripening time of cheeses, ensuring the effective inactivation of microorganisms and thus improving safety [5,6]. For example, studies performed with *Torta del Casar* cheese [5,6] demonstrated that HPP treatment of 600 MPa/5 min at 60 days of ripening caused 2.3 and 2.5 log cycle reductions in *Enterobacteriaceae* and *Pseudomonas* spp. viable cells numbers, respectively [5]. In what concerns *Serra da Estrela* cheese, HPP treatment of 450 and 600 MPa for 6 min enabled lactic acid bacteria (LAB) viable cell numbers to be kept at similar levels to control cheeses and only minor changes were verified among physicochemical parameters (data submitted). Both studies also showed interesting results in terms of proteolytic indexes; HPP deaccelerated proteolysis kept the ripening extension index closer to that of control cheeses and, therefore, closer to the optimum organoleptic attributes (data submitted) and delayed the over-ripening during refrigerated storage up to commercialization [5]. 

HPP treatment requires products to be packed in waterproof packages (usually polyamide–polyethylene plastic films for solid foods such as cheese) prior to processing. However, consumers still prefer wrapping paper package systems without vacuum for this type of traditional cheese (raw milk cheeses). Nevertheless, foods cannot be processed by HPP if wrapped only in paper. 

The feasibility of HPP as a non-thermal treatment for raw milk cheese pasteurization has been studied, where once processed, the different cheese types were stored under a vacuum [5,6]. However, it was verified that the cheeses’ rind became whiter over time, a negative aspect that could have been avoided if access to oxygen was enabled by using oxygen-permeable packages [5].

Based on this, two packaging systems for storage of *Serra da Estrela* raw milk cheeses, after HPP application at an optimum organoleptic time, were assessed: (i) vacuum packaging in polyamide–polyethylene plastic bag and (ii) non-vacuum packed in greaseproof wrapping paper (cheeses were removed from the plastic bag after HPP and kept wrapped in paper for the study). To the best of our knowledge, no other study is available in the literature on this specific research topic. 

## 2. Materials and Methods

### 2.1. Milk Supply and Cheese Manufacture 

One hundred and fifty liters of raw ewe milk (from two farms in the *Serra da Estrela* cheese production region, Portugal) were collected and kept in a refrigerated reservoir until further use. Prior to sampling, milk was well mixed to ensure a homogeneous sampling. Two batches of *Serra da Estrela* cheese were produced at the dairy, one in the morning and the other in the early afternoon, according to the PDO procedures [1,2]. The resulting 56 cheeses (of about 500 g each) were ripened for 45 days according to the PDO practices (first 15 days at 11 ± 2 °C and 75% HR and then at 9 ± 2 °C and 90% HR) [2] in order to reach the optimum organoleptic quality. Upon 45 days of ripening at the dairy, half of the cheeses (n = 28) were wrapped in paper (50 g m^−2^ white Kraft paper plus 10 g m^−2^ low-density polyethylene (LDPE) from Seilimp, Oliveira do Hospital, Portugal) and all 56 cheeses were then vacuum packaged (vacuum packaging machine HenkoVac E-193, Albipack, Aveiro, Portugal) in heat-sealed polyamide–polyethylene film (PA-PE, 90 μm, 16 μm PA and 74 μm PE, IdeiaPack—Comércio de Embalagens, Lda, Viseu, Portugal). 

### 2.2. High-Pressure Processing

Twenty-eight cheeses were treated by HPP treatments in a 55-L capacity industrial scale high-pressure equipment (model 55, Hiperbaric, Burgos, Spain) at 525 MPa for 6 min (this condition was selected based on previous results obtained for *Serra da Estrela* cheese pasteurization (data submitted). The initial temperature of the water used as transmitting fluid was 8 °C, depressurization took less than 2 s, and the two cheese batches were processed in two different high-pressure processing cycles. 

### 2.3. Samples Identification and Sampling

Several samples were collected for chemical characterization, including the refrigerated milk used for cheese manufacture, both morning and early afternoon batches, and the resulting curd upon squeezing (1.5 h after starting milk coagulation).

The fifty-six cheeses produced were divided equally into two groups: HPP-processed (n = 28) and unprocessed cheeses (control cheeses, 28). In order to study the effect of the packaging system, half of the unprocessed cheeses (Ch_C_) were kept under vacuum in polyamide–polyethylene plastic film (Ch_C_+V, 14 cheeses) and the other 14 cheeses were kept wrapped in a greaseproof paper without vacuum (Ch_C_+P); similarly, the HPP-treated cheeses (Ch_P_) were kept half (14 cheeses) under vacuum in polyamide–polyethylene plastic film (Ch_P_+V) and the other half (14 cheeses) were removed from the plastic bag and kept without vacuum wrapped in greaseproof paper (Ch_P_+P). All 56 cheeses were kept refrigerated at 4 °C for 10 months, having the non-vacuum paper-wrapped cheeses being washed due to visible mold development after 3- and 6-month storage. The washing step was carried out according to the PDO mandatory procedures (water and a wash brush), except that sterilized water was used and the washing took place under aseptic conditions in a laminar flow cabinet to avoid microbial contamination; it was important to keep the non-vacuum paper wrapped cheeses under conditions as similar as possible to those of the vacuum-packed cheeses, in what concerns the prevention of microbial contamination. At each storage time (0, 3, 6, and 10 months), aliquots of each cheese (≈35 g per sample) were stored at −80 °C until physicochemical analyses were carried out.

### 2.4. Microbiological Analyses

At each sampling point, 8 cheeses were cut in half, and a thin slice was cut through the innermost, the intermediate, and the outermost layers of cheese. The rind was removed, and the three portions obtained were mixed to obtain a single 10 g cheese sample. This sample was aseptically handled and homogenized for 4 min using a 2% (*w v*^−1^) aqueous sodium citrate solution as extraction buffer in a Stomacher Lab-Blender 400 (Milano, Italy). Aliquots of 1.5 mL were then taken and decimally diluted in 13.5 mL of sterile 0.1% (*w v*^−1^) aqueous peptone; decimal dilutions were subsequently prepared, and these were then plated in triplicate. The following microbial groups were enumerated using the pour plate method: *Enterobacteriaceae* on violet-red bile dextrose agar (VRBDA from Merck, Darmstadt, Germany); coliforms and *E. coli* on chromocult coliform agar (CCA from Merck), both incubated at 37 °C for 24 h. The Miles and Misra technique [7] was used for enumeration of: *Enterococcus* spp. on kanamycin aesculin azide agar base (KAAA, Oxoid, Basingstoke, UK) and incubated at 37 °C for 24 h; *Lactobacillus* spp. on Man, Rogosa and Sharpe (MRS, Merck) and incubated at 30 °C for 3 d; *Lactococcus* spp. on M17 (Liofilchem, Roseto degli Abruzzi Térano, Italy) and incubated at 30 °C for 3 d; and *Bacillus* spp. on HiChrome (Fluka, Buchf, India) and incubated at 30 °C for 2 d; total aerobic mesophilic microorganisms on plate count agar (PCA, Merck) and incubated at 30 °C for 3 d; total anaerobic microorganisms on PCA and incubated at 37 °C for 2 d in anaerobic jars (Merck) with Merck Anaerocult A (Merck); total psychotropic microorganisms on PCA and incubated at 20 °C for 5 d; yeasts and moulds on rose-bengal chloramphenicol agar (RBCA, Merck) and incubated at 25 °C for 5 d; *Staphylococcus* spp. on Baird-Parker agar (BPA, Merck) with egg yolk tellurite emulsion (Liofilchem) and incubated at 37 °C for 2 d; *Listeria* spp. on PALCAM agar selective agar base (Liofilchem), with selective supplement for PALCAM (Liofilchem) and incubated at 37 °C for 2 d; and *Pseudomonas* spp. on pseudomonas agar base (PAB from Liofilchem) with glycerol and pseudomonas CFC supplement (CFC, Liofilchem) and incubated at 30 °C for 2 d. Petri dishes containing 30–300 and 10–100 colony forming units (cfu) were selected for viable cell counts quantification for the pour plate and Miles and Misra, respectively. The results were converted into logarithmic decimals of the number of cfu *per* g of cheese sample, and the values were considered below the limit of quantification of 2.0 log cfu g^−1^ for the pour plate technique and 3.0 log cfu g^−1^ for Miles and Misra technique. Less than 1 log cfu g^−1^ was considered for milk samples due to direct liquid sample plating.

### 2.5. Physicochemical Analyses 

The pH of the cheeses was measured at room temperature, randomly on the cheese, using a penetration pH meter (Testo 205, Testo, Inc., West Chester, PA, USA). The titratable acidity was determined according to the AOAC 920.124 procedure [8], using an automatic titrator with a pH meter (Crison—Titromatic 1S with pH electrode 50 14, Barcelona, Spain) by titration to a pH end-point of 8.9. Moisture content was determined by drying approximately 2 g of cheese to a constant weight (ca. 24 h) at 105 °C using laboratory oven-drying equipment (Venticell, MMM Medcenter Einrichtungen GmbH, Munich, Germany). Physicochemical analyses were performed in triplicate per cheese sample. 

### 2.6. Color

Color parameters were measured using a Minolta Konica CM 2300d (Konica MinoltaCM 2300d, Osaka, Japan) at room temperature. The color parameters were recorded using the CIE Lab system and directly computed through the original SpectraMagic NX software (Konica Minolta, Osaka, Japan), according to the International Commission on Illumination regulations. Cheese samples were kept 1 h at room temperature before measurements. Measurements were performed on each of six random spots *per* cheese sample (surface and core).

### 2.7. Proteolysis

Proteolysis was monitored during storage by measuring the amount of nitrogen content of different cheese fractions by the micro-Kjeldahl method [9,10], using a Kjeltec system with a 2012 digester and a 1002 distilling unit (Tecator, Hoganas, Sweden). Cheese fractions soluble in water (WSN), in 12% (*w v*^−1^) trichloroacetic acid (TCA), and in 5% (*w v*^−1^) phosphotungstic acid (PTA) were prepared, and the nitrogen content measured according to Macedo and Malcata (1997) [11]. The analyses were run in duplicate per cheese. The quantity of nitrogen soluble in water, in 12% TCA and 5% PTA was expressed as per unit mass of total nitrogen content (TN) and will be denoted hereafter as WSN TN^−1^ as ripening extension index, TCA TN^−1^ as ripening depth index, and PTA TN^−1^ as a free amino acid index.

### 2.8. Instrumental Texture Profile Analysis (TPA)

Each cheese was kept at room temperature (18–22 °C) for 2 h before analysis. Random cylinders of cheese (18 mm diameter) were taken with a cork borer inserted vertically through the cheeses from their top surface, crossing from side to side, and 3 mm were cut off from each side, corresponding to the cheese rind. The texture was analyzed by TPA tests consisting of two sequential penetration cycles of 10 mm penetration at a rate of 0.80 mm s^−1^, separated by a rest period of 10 s, using a texture analyzer (TA-Hdi, Stable Micro Systems, Godalming, UK) connected to a 2 mm diameter probe. The tests generated a force–time curve, from which hardness (N), consistency (N s^−1^), adhesiveness (N s^−1^), cohesiveness, and gumminess (N) were calculated [12]. All analyses were performed in sextuplicate per cheese. 

### 2.9. Sensorial Evaluation

The sensory evaluation of cheeses was carried out by ten trained panelists from the Faculty of Biotechnology (CBQF, Porto, Portugal). Sensory sessions took place at the ISO 8589:2007 compliant sensory evaluation laboratory of CBQF, equipped with white fluorescent lighting (6500 K). Analyses were carried out at room temperature (18–22 °C).

On each testing day, the cheeses were removed from refrigeration about 1 h prior to evaluation and kept at room temperature. The rind was removed in one half of each cheese which was then cut into slices with 0.7 cm of thickness of the outermost cheese layer. Mineral water and Granny Smith apple pieces were provided to the panel members to cleanse their palates between samples. Panel sessions were always held mid-morning. Qualtrics online questionnaires (Qualtrics, LLC, Provo, UT, USA) were used. 

Three sensory evaluation sessions were carried out at each sampling date: 0 months, 3 months, and 6 months, and only one session at 10 months. Firstly, two paired comparison tests were carried out in two sessions: in the first session, panelists compared Ch_P_+V with Ch_C_+V to ascertain possible differences caused by the effect of HPP on cheeses vacuum-packed packaging in plastic film; in the second session, Ch_P_+P was compared with Ch_C_+P to assess possible differences caused by the effect of HPP on cheeses stored wrapped in paper without vacuum. The attribute difference-from-control method was used to compare the magnitude of difference between the intensity of each evaluated attribute of the HPP cheeses relative to the control cheeses, using a bipolar anchored continuous scale (−10 = much less intense …, 0 = no difference, 10 = much more intense …). The attributes evaluated by panelists were: rind appearance (tonality from much lighter to much darker, homogeneity and defects), paste appearance (color from much lighter to much darker and consistency from much more fluid to much firmer), odor (lactic, acid, animal/stable and short-chain fatty acids (SCFA)/vomit from much less intense to much more intense), texture (consistency from much softer to much harder, unctuosity and friability from much less to much more), taste (salty, acid, and bitter from much less intense to much more intense) and after-taste (much less intense to much more intense). For rind evaluation, one-half cheese was presented to panelists, and for the remaining attributes, cheese slices were provided to panelists in Petri dishes. In each of the two sessions, one sample of the control cheese was presented to panelists identified as such; a second sample of the control cheese (blind control sample) plus a sample of the HPP cheese were presented to panelists coded with three-digit random numbers. In the third session, a rating test was used to evaluate the intensity of the abovementioned attributes for the four cheeses. A continuous anchored scale was used (0 = absent, 10 = strong), and samples were coded with three-digit random numbers. In this third evaluation session, the objective was to evaluate the effect of HPP and the type of storage packaging system on cheese quality. 

### 2.10. Statistical Analyses

Analysis of variance (ANOVA) was performed to establish the effect of different processing/packaging systems conditions, the effect of storage, and both. Significant differences were investigated using a post hoc test—Bonferroni procedure, with the significance assigned at *p* < 0.05. Data without normal distribution were analyzed by the Kruskal–Wallis test. Attribute difference-from-control test sensory data were analyzed by paired t-student comparison between Ch_P_+V vs. Ch_C_+V and Ch_P_+P vs. Ch_C_+P, with the significance assigned at *p* < 0.05; when the distribution of the differences between the control and treated cheeses did not show a normal distribution, the non-parametric test Wilcoxon was applied. Sensory rating test data were analyzed by one-way ANOVA, and a Tukey’s post hoc test was applied to compare the mean values of attributes for each storage time. SPSS software version 26.0 (Armonk, NY: IBM Corp) was used for the statistical analysis. 

## 3. Results and Discussion

### 3.1. Microbial Composition of Milk and Fresh Curd

In milk samples, lactobacilli and lactococci revealed a microbial load of 4.23 and 5.72 log cfu mL^−1^, respectively, and total aerobic mesophilic bacteria viable cell numbers of 5.73 log cfu mL^−1^. *Enterobacteriaceae*, coliforms, and enterococci viable cell numbers were found at similar levels, namely, 2.98, 2.60, and 2.42 log cfu mL^−1^, respectively. Staphylococci and *pseudomonas* spp. were detected at 4.21 and 3.36 log cfu mL^−1^, respectively. In the curd, viable cell numbers of 5.21, 5.70, 4.13, and 6.36 log cfu g^−1^ were measured for lactobacilli, lactococci, enterococci, and total aerobic mesophiles, respectively, which are values close to those already reported for this type of curd [1], while *Enterobacteriaceae* and coliforms were quantified at 4.96 and 4.28 log cfu g^−1^ (ca. 2 log cycle increase in comparison to milk microbial load), respectively, being ca. 2 log cycles below that reported in the literature [1]. Viable cell numbers of staphylococci remained fairly stable at 4.13 log cfu g^−1^ and *Pseudomonas* spp. increased ca. 1 log cycle to viable cell numbers of 4.73 log cfu g^−1^.

### 3.2. Changes in Microbial Composition Induced by HPP and Packaging System

Figure 1 shows the viable cell numbers of the different microbial groups found in HPP-treated and non-treated *Serra da Estrela* cheeses stored for 10 months at 4 °C under a vacuum in plastic film or wrapped in paper without vacuum. In general, viable cell numbers were not significantly affected (*p* ≥ 0.05) by the packaging system type, but they were affected by HPP treatment (*p* < 0.05). Lactobacilli and lactococci (Figure 1A,B) were found at a similar order of magnitude independently of the cheese treatment and packaging system; in Ch_C_+V cheeses, viable cell numbers were found at 8.37 and 8.29 log cfu g^−1^, respectively, at the beginning of storage (0 months), and these remained relatively constant (*p* ≥ 0.05) throughout the 10 months of storage (exception for lactococci viable cell numbers that increased to 9.30 log cfu g^−1^ at 3 months of storage (*p* < 0.05)). Similar values were reported in the literature, never below the 8 log cfu g^−1^ at 30–60 days of ripening [13,14]. The HPP treatment caused a significant reduction in lactobacilli and lactococci viable cell numbers (*p* < 0.001) of about 3.8 and 2.5 log cycles, respectively, values close to those previously obtained for *Serra da Estrela* cheese (450 and 600 MPa/6 min) (submitted data) and in Casar cheese (400 and 600 MPa/5 min) [15]. At each sampling point, no significant differences in viable cell numbers between cheeses vacuum-packed in plastic film and non-vacuum wrapped in paper were observed (*p* ≥ 0.05). Enterococci counts revealed a behavior similar to that of LAB (Figure 1C), having the Ch_P_+V and Ch_P_+P cheeses showed a significant decrease of 1.00 and 1.12 log cycles in enterococci viable cell numbers in comparison to Ch_C_+V and Ch_C_+P cheeses (*p* < 0.001), respectively. A higher reduction of about 2–3 log cfu g^−1^ was verified for HPP (400 or 600 MPa/5 min) Casar cheeses [15]. Total aerobic mesophiles revealed viable cell numbers of 8.61 log cfu g^−1^ in Ch_C_+V cheeses at 0 months of storage, with no further variations along storage time (*p* ≥ 0.05), as can be observed in Figure 1D. The packaging system type also had no significant effect on total aerobic mesophiles (*p* ≥ 0.05), while HPP led to a significant decrease of about 2.6 log cycles of this group of microorganisms (*p* < 0.01). In the literature, lower reductions in viable cell numbers (0.88–1.33 log units) were achieved for both lower and higher pressure intensity treatments (400 or 600 MPa/5 min) of Casar cheeses treated at 35 days of ripening [15]. Nevertheless, the results in the present work are in agreement with values previously reported for total aerobic mesophiles in HPP *Serra da Estrela* cheese (2.4–5.3 log cycle reductions) (submitted data). For anaerobic and psychotropic microorganisms (Figure 1E,F), a behavior similar to that of total aerobic mesophiles was found.

The coliforms total counts in Ch_C_+V cheeses at 0 months were in agreement with the 5–6 log cfu g^−1^ reported at 35 days of ripening by Macedo et al. (1995, 1996) [3,16], and these remained stable throughout storage (*p* ≥ 0.05) (Figure 1G). HPP caused 1 log cycle reduction at 0 months storage, which intensified along storage, making viable cell numbers drop to values closer to 3 log cfu g^−1^; a higher reduction (>3.5 log cycle units) was reported for similar HPP treatments by Arqués et al. (2006) [17] and Calzada et al. (2013) [15].

*Enterobacteriaceae*, *pseudomonas* spp., and *E. coli* viable cell numbers were reduced to below the quantification limit by the HPP treatment (Figure 1H). In contrast, HPP-treated (200 or 600 MPa/5 or 20 min) Casar cheeses, kept under vacuum for 180 days, revealed *pseudomonas* counts in the cheese rind [5]. The HPP treatment caused staphylococci viable cell numbers to reduce to levels of ~3.8 log cfu g^−1^, below the established limit of 10^5^ cfu g^−1^ [18] (Figure 1I). *Listeria* spp. was below the detection limit in all cases.

Yeasts and molds viable cell numbers in Ch_C_+V cheeses reached between 4.07 and 6.48 log cfu g^−1^ over the 10 months of storage, as can be seen in (Figure 1J). HPP caused significant reductions of >3.4 log cycles to below the quantification limit; however, in this case, the packaging system played an important role. In the case of Ch_P_+P cheeses, this effect vanished with time since yeasts and molds proliferated in the rind, which made the cheeses require washing.

As far as the authors are aware, there are no reports in the literature concerning the impact of storage on the microbial quality of non-vacuum paper-wrapped cheeses. On the other hand, in the case of vacuum-packed ewe cheese storage, HPP treatments (600 MPa/5 min) caused > 2.1 log units reduction of molds, to below the quantification limit at 180 days [5].

Overall, Ch_C_+V cheeses revealed viable cell numbers closer to those already reported for *Serra da Estrela* cheese. HPP caused microbial reductions in the range reported in the literature for vacuum-packaged and stored cheeses. In general, the packaging system type caused no significant differences in viable cell numbers, exception for yeasts and molds, which showed growth in non-vacuum paper-wrapped cheeses, but not in vacuum-packed plastic film cheeses, independently of non-treated or HPP treatment.

### 3.3. Changes in Physicochemical Characteristics

Despite the natural decrease of the moisture content during storage, neither the HPP treatment nor the packaging system (vacuum vs. non-vacuum/paper vs. plastic film) influenced moisture and protein contents significantly (Table 1). The measured values were found within the ranges reported in the literature for this cheese, i.e., 40–48 and 14–26%, respectively [19,20]. Similarly, Delgado et al. (2015) [6] demonstrated that HPP treatments (200 or 600 MPa/5 or 20 min) did not have an impact on the moisture content of ripened ewes’ cheese.

At the beginning of storage (0 months), the Ch_C_+V cheeses had pH values of 5.24, similar to the other three types (Ch_C_+P, Ch_P_+V, and Ch_P_+P) of cheeses (*p* ≥ 0.05). As storage time increased, so did pH values of Ch_C_+P cheeses, in particular, reaching a significantly different pH value of 6.69 at 6 months of storage (*p* < 0.001). Interestingly, non-treated and HPP-treated paper-wrapped cheeses, Ch_C_+P and Ch_P_+P, were the cheeses that revealed the highest increase in pH values over a 6-month storage period; in fact, both cheese types showed pH values higher than those normally reported in the literature for this cheese, i.e., 4.82–5.66, which is aligned with the higher proteolytic indices discussed below—the release of free amino acids will increase the cheese pH value [13,20,21,22]. Similarly to a previous study, Ch_C_+V cheeses revealed the highest TA, probably due to higher endogenous microbial counts and, consequently, a higher metabolic activity [2]. Non-treated and HPP-treated paper-wrapped cheeses maintained TA values relatively stable over 6 months of storage. We should highlight that the lack of reports in the literature concerning *Serra da Estrela* cheese storage, as well as that of other cheeses when wrapped in greaseproof paper, does not enable further comparisons.

### 3.4. Changes in Proteolytic Indexes

At the beginning of storage (0 months), the ripening extension and depth indexes were not significantly affected either by HPP treatment or by packaging system type, being closer to 25–26% and 6–7%, respectively (*p* ≥ 0.05) (Figure 2A,B). These values are slightly lower than those reported in the literature for *Serra da Estrela* cheese of about 29–37% and 6–13% for WSN TN^−1^ and TCA TN^−1^ ratios, respectively [22,23]. However, during storage, a significant increase of the WSN TN^−1^ and TCA TN^−1^ ratios occurred, reaching 42.8 and 10.2%, respectively in Ch_C_+P cheeses at 6 months of storage (*p* < 0.001). These higher proteolytic indexes in non-vacuum paper-wrapped cheeses were reflected in the higher pH values, as discussed above; higher pH values may favor proteolysis since the optimum pH for most proteinases and peptidases is close to 7 [24]. These results seem to indicate a more intense post-ripening metabolic activity in Ch_C_+P cheeses.

On the other hand, the HPP-treated cheeses, independently of the packaging system, Ch_P_+P and Ch_P_+V, maintained the WSN TN^−1^ (27 and 26%, respectively) and the TCA TN^−1^ (8 and 7%, respectively) values relatively stable over 6 months of storage, values which were close to those obtained at the starting point (0 months) for the Ch_C_+V cheeses, i.e., 25 and 6%, respectively (*p* ≥ 0.05). These results tend to indicate that the HPP treatment and the keeping of the cheeses under vacuum led to a reduction in the production of medium- and small-size peptides included in the WSN and TCA fractions, respectively. Previous studies have shown that more intense HPP treatments at 600 MPa/6 min for *Serra da Estrela* cheeses (submitted data) and at 600 MPa/20 min for Torta del Casar cheeses [6] reduce proteolysis development during 500 and 240 days of storage under vacuum, confirming the possibility of HPP treatment to keep the ideal ripening characteristics during extended storage periods. The PTA TN^−1^ ratio was (unexpectedly) significantly affected by the interaction between HPP treatment and the vacuum packaging system used at 0 months (*p* < 0.05); HPP treatment lowered the PTA TN^−1^ ratio of Ch_P_+V cheeses. Regardless of the effect of HPP on the free amino acid index of *Serra da Estrela* cheeses, at 6 and 10 months of storage, no significant changes in the PTA TN^−1^ ratios (*p* ≥ 0.05) were verified for all 4 cheese types.

### 3.5. Color

No significant effect of the HPP treatment and/or packaging system type on the cheese surface *L** and *a** color parameters and cheese core *L** color parameter (measured the lightness from black (0) to white (100) (*p* ≥ 0.05) was observed (Table 2). On the other hand, the *b** color parameter of the cheese core, which measures the blue (−) to yellow (+) color, was significantly higher for Ch_P_+V, Ch_C_+P, and Ch_P_+P than for Ch_C_+V cheeses (*p* < 0.001), indicating these former cheeses as being yellower. Along the storage period, the *a** color parameter of the cheese surface and core, which measures from green (−) to red (+) color, was maintained constant in all 4 cheese types (*p* ≥ 0.05). On the other hand, the surface of vacuum-packaged plastic film cheeses became lighter, and the surface of non-vacuum paper-wrapped cheeses became yellower (higher *L** and *b** parameters, respectively) (*p* < 0.05), having the latter gained a very heterogeneous and non-characteristic color at 6 months, as can be observed in (Figure 3), and so the color of these cheeses was not quantified. Delgado et al. (2013) [25] also measured higher *L** and *b** color parameters on Ibores cheese HPP (400 or 600 MPa/7 min) after 1 month under vacuum. According to Voigt et al. (2010) [26], the color changes induced by HPP can be related to the effect of processing on hydrophobic bonds between casein molecules, which changes the light-scattering of the HPP-treated cheese; or the HPP treatment might involve the release of pigments by the cheese molds. It has been pointed out in the literature that vacuum storage causes the rind to become whitish [5,25]. In order to try and overcome such limitations, the present work studied an alternative packaging system (non-vacuum and paper wrapping), which hypothetically could keep the achieved HPP advantages and avoid the changes in the color of the rind. However, Ch_C_+P and Ch_P_+P cheeses rind revealed a high total color variation of 8.26 and 10.46, respectively by 3 months of storage (relative to the beginning of storage), possibly due to higher oxygen availability in the case of Ch_C_+P and Ch_P_+P cheeses. For a short storage period (less than 3 months), the non-vacuum + greaseproof wrapping paper could be an interesting way to package cheese, but for longer storage periods, the vacuum packaging in polyamide–polyethylene plastic film method is preferable. Curiously, when Ch_C_+V and Ch_P_+V cheeses were unpacked and kept for some days at atmospheric pressure, they became yellower.

### 3.6. Changes in Textural Properties

HPP showed no significant effect on cheese textural properties at 0 months of storage (comparison of Ch_C_+V with Ch_P_+V and Ch_C_+P with Ch_P_+P cheeses, *p* ≥ 0.05) (Table 3). On the other hand, cheeses vacuum-packed in plastic film revealed significantly lower hardness (0.18–0.27 N vs. 0.56–0.58 N) (*p* < 0.001), lower consistency (1.4–1.7 N s^−1^ vs. 4.8–5.1 N s^−1^) (*p* < 0.01), lower adhesiveness (0.3–0.6 N s^−1^ vs. 1.3–1.4 N s^−1^) (*p* < 0.001) and higher cohesiveness (5.5–6.6 vs. 2.6–2.7) (*p* < 0.01) than non-vacuum paper-wrapped cheeses at 0 months of storage. During the first 6 months of storage, both Ch_C_+V and Ch_P_+V cheeses revealed an increase in hardness and consistency features (to 0.49–0.59 N and 4.1–5.1 N s^−1^, respectively) (*p* < 0.01) and a decrease in cohesiveness values (to 2.3–4.2) (*p* < 0.05); no significant differences in these parameters were reported among Ch_C_+V and Ch_P_+V (*p* ≥ 0.05) in the first 3 months. The increase in these texture parameters could be related to the loss of moisture throughout refrigerated storage (Table 1). This behavior was also verified in a previous study in HPP-treated (at 600 MPa/6 min) and control cheeses vacuum-packed in plastic for 15 months (submitted data). A study performed on Torta del Casar cheeses revealed a similar behavior, an increase in the consistency over 6 months under vacuum, without significant differences between vacuum-packaged control and HPP (600 MPa/5 min) treated cheeses, which could be related to the loss of moisture during storage [6]. A study on Ibores raw goat milk cheese showed similar behavior for cheeses kept for 30 days under vacuum; HPP-treated cheeses (600 MPa/7 min) revealed higher hardness and lower adhesiveness than control ones [25].

### 3.7. Changes in Sensorial Attributes

Paired comparison between Ch_P_ and Ch_C_ samples at 0 months storage revealed significant differences between non-vacuum paper-wrapped control and HPP cheeses for rind defects and paste appearance (Table 4), with Ch_P_+P having been judged to present fewer rind defects than Ch_C_+P (*p* < 0.05), a paste with lighter color tone and a less firm consistency (*p* < 0.05). These results were confirmed by results obtained in the rating test in which the four cheese samples were evaluated (Table 5). No significant differences in odor attributes were found at this storage time. As discussed in the previous section, TPA analysis did not reveal significant differences in hardness among Ch_C_+V vs. Ch_P_+V and Ch_C_+P vs. Ch_P_+P cheeses at 0 months storage (Table 3), which might be related to the particularities of sensorial and TPA analysis, such as sensitivity. At 3 months of storage, significant differences between the HPP-treated and the control cheeses vacuum packed in plastic film were found for most of the evaluated appearance, texture, and flavor attributes; between the HPP-treated cheeses and the control cheese stored at non-vacuum paper-wrapped conditions differences were found for rind appearance and texture attributes. In relation to appearance, Ch_P_+V cheeses were judged to have a significantly darker rind and paste color tone, with a less homogenous rind and a firmer paste consistency (*p* < 0.05) than Ch_C_+V cheeses. The rind appearance of Ch_P_+P cheeses was revealed to have a darker color tone and fewer defects (*p* < 0.01) (Table 4). The panel rated the Ch_P_+P cheeses with a more intense lactic odor than the Ch_C_+P cheeses. The HPP-treated cheeses were judged to have a firmer paste than the controls, as verified in a previous study (data submitted). Nevertheless, instrumental measured texture revealed only significantly lower consistency values for Ch_C_+P cheeses (*p* < 0.001). Delgado et al. (2013) [25] verified that odor and flavor intensity were not affected by HPP (600 MPa/7 min) for Ibores raw goat milk cheeses evaluated 1 month after treatment.

At 6 months, the panel attributed lighter rind and paste color tone for cheeses vacuum-packed in plastic film (Figure 3 and Table 5), particularly for Ch_C_+V cheeses, being in agreement with instrumental color evaluation. Similarly, a darker yellow color appearance was attributed to Ibores raw goat milk cheeses 1 month after HPP treatment (600 MPa/7 min) than control ones [25]. Ch_C_+P and Ch_P_+P cheeses at 6 months revealed significantly lower (*p* < 0.05) acid odor and flavor (Table 5). These results indicated that Ch_C_+P and Ch_P_+P cheeses were clearly outside of the expected characteristics for this type of cheese, and so the evaluation was ended at this point for these cheeses.

At 10 months, three attributes (rind tonality, color paste, and texture consistency) revealed that HPP had an effect on Ch_P_+V cheeses, but with a positive effect, where the cheese rind and paste color tone became darker (in accordance with lower *L** color parameter). Due to the absence of reports in the literature on sensorial analysis for longer cheese storage times, the comparison of results is not straightforward. Nevertheless, Casar raw ewe milk [27,28] and raw cows’ milk cheeses [29] were HPP treated (400 or 600 MPa/5 min) at 21 or 35 days of ripening; both were unpacked for ripening during the 2 months, followed by storage at 4 °C for 6 months. Along storage, raw ewe milk cheeses processed by HPP revealed significantly lower odor intensity but significantly higher odor quality, as well as lower putrid and rancid odors than control ones [27]; the flavor intensity was slightly lower, but the flavor quality was significantly higher than that of the control cheeses [28]. In raw cow milk cheeses, HPP-treated and control cheeses showed similar flavor intensity and quality, but the HPP cheeses revealed a more pronounced bitter flavor than the controls [29], in contrast to the results presented in this work. Overall, from the sensorial point of view, it can be concluded that for longer storage times, cheese packaging in plastic film under vacuum is preferable, while for shorter periods, a non-vacuum paper wrapping system is also a viable option.

## 4. Conclusions

Control cheeses revealed viable cell numbers of different microbial groups closer to values commonly reported for *Serra da Estrela* cheese. HPP treatment caused microbial reductions in the range of those reported in the literature, being more pronounced for gram-negative bacteria such as *Enterobacteriaceae*, *Pseudomonas* spp., and *E. coli*, which were reduced to below the quantification limit, while for lactobacilli, lactococci, enterococci, and total aerobic mesophilic microorganisms, a reduction of about 1–3 log cycle units was verified. In general, the packaging system did not have a significant impact on viable cell numbers. However, yeasts and molds grew more (>5 log cfu g^−1^) in non-vacuum paper-wrapped cheeses in comparison to vacuum plastic film packaged cheeses, whose presence was also perceived by the sensorial panel, while the rind of the latter cheeses became whitish.

Ripening extension and depth indexes were maintained relatively constant for long storage periods in HPP-treated cheeses, independently of the packaging system type, leading to a harder texture. Sensorial analysis indicated that HPP cheeses vacuum packaged in the plastic film kept the main attributes for up to 10 months, while this was verified for only 3 months for non-vacuum paper-wrapped cheeses.

Overall, these results allow concluding that HPP had a positive effect on the maintenance of raw ewe milk *Serra da Estrela* cheese safety and quality characteristics and that storage under vacuum in plastic films is more adequate than non-vacuum paper wrapping.

## Figures and Tables

**Figure 1 foods-12-01935-f001:**
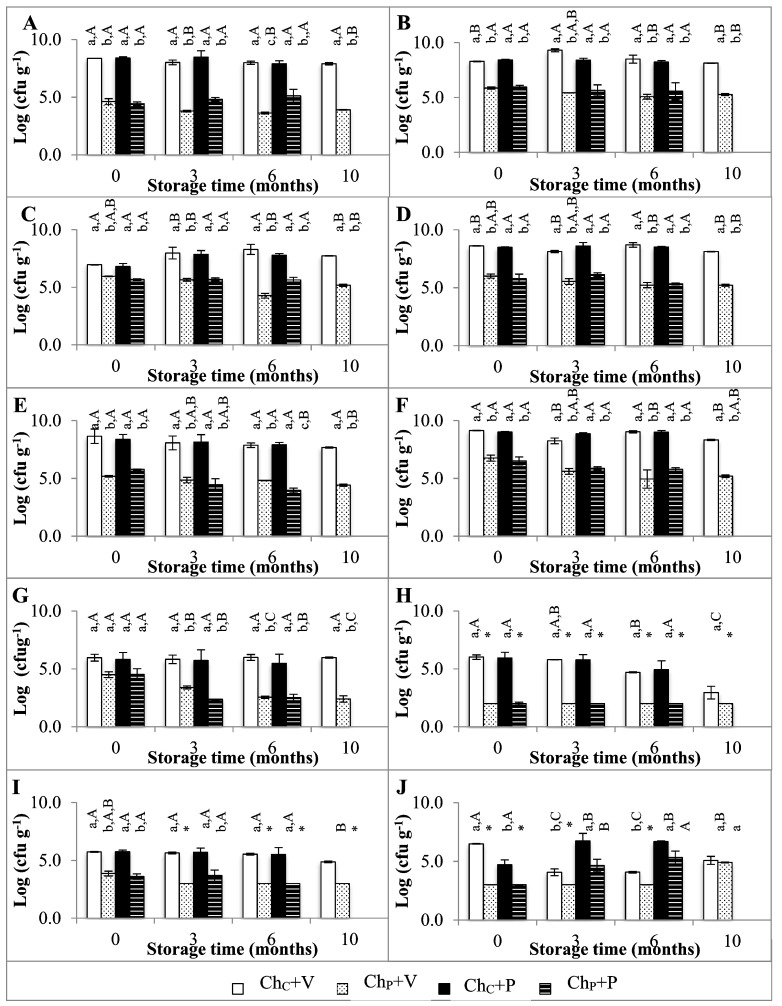
(**A**) Lactobacilli, (**B**) lactococci, (**C**) enterococci, (**D**) total aerobic, (**E**) anaerobic, (**F**) psychrotrophic, (**G**) total coliforms, (**H**) *Enterobacteriaceae*, (**I**) staphylococci, (**J**) yeasts and molds counts in *Serra da Estrela* cheese at 0, 3, 6, and 10 months of refrigerated storage (control cheeses stored under vacuum in plastic film (Ch_C_+V) or wrapped in paper without vacuum (Ch_C_+P) and HPP-treated cheeses stored under vacuum in plastic film (Ch_P_+V) or wrapped in paper without vacuum (Ch_P_+P). * means below the quantification limit. Different non-capital letters (a, b, c) indicate statistically significant differences between the same storage time, while different capital letters (A, B, C) indicate statistically significant differences among the same condition (*p* < 0.05).

**Figure 2 foods-12-01935-f002:**
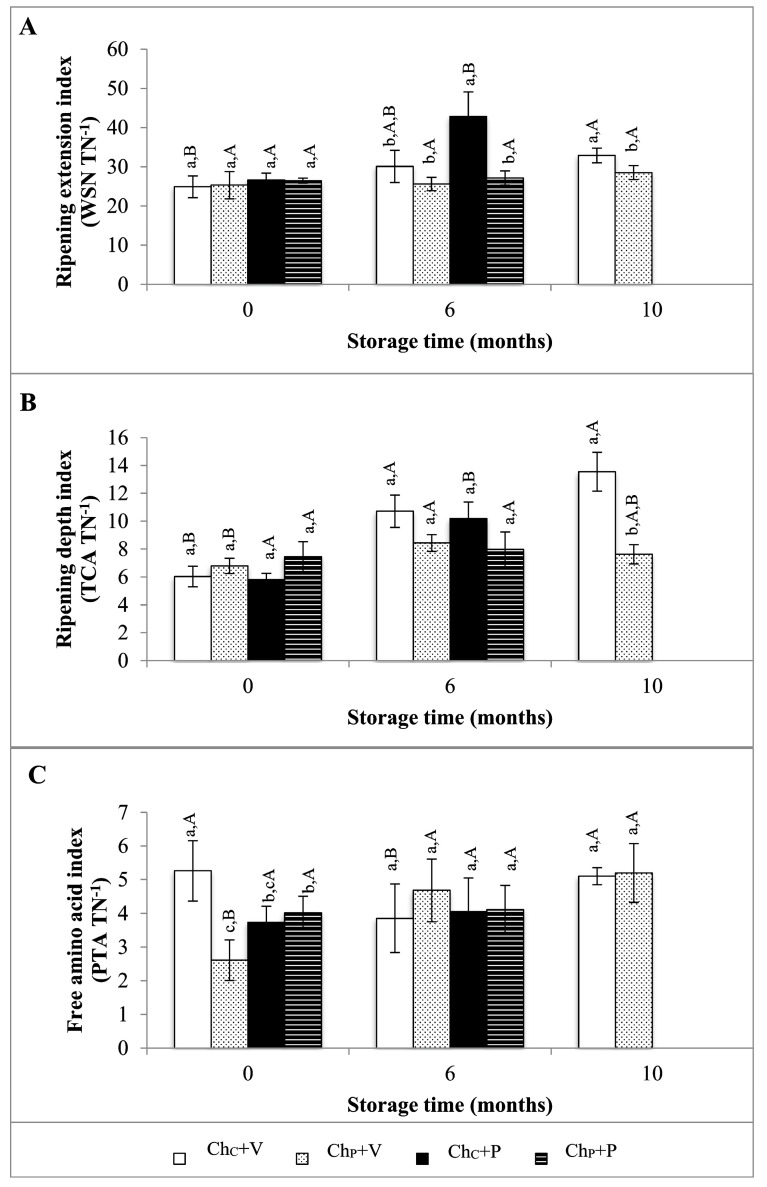
Evolution of (**A**) ripening extension index (WSN TN^−1^), (**B**) ripening depth index (TCA TN^−1^), and (**C**) free amino acid index (TCA TN^−1^) of *Serra da Estrela* control cheeses stored under vacuum in plastic film (Ch_C_+V) or wrapped in paper without vacuum (Ch_C_+P) and HPP-treated cheeses stored under vacuum in plastic film (Ch_P_+V) or wrapped in paper without vacuum (Ch_P_+P) at 0, 6, and 10 months of storage. Different non-capital letters (a, b, c) indicate statistically significant differences between the same storage time, while different capital letters (A, B, C) indicate statistically significant differences among the same condition (*p* < 0.05).

**Figure 3 foods-12-01935-f003:**
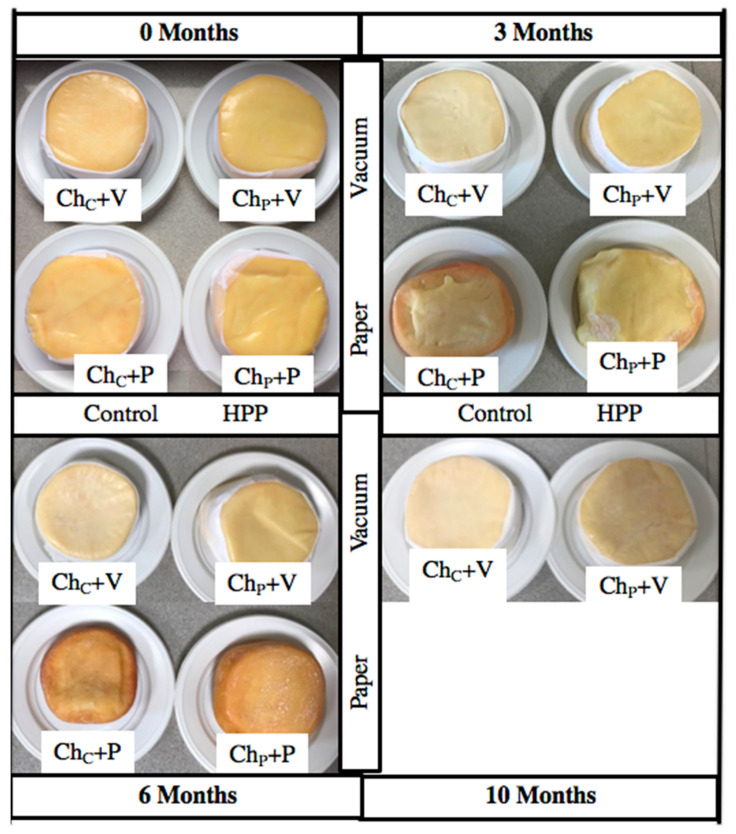
Visual appearance of *Serra da Estrela* control cheeses stored under vacuum in plastic film (Ch_C_+V) or wrapped in paper without vacuum (Ch_C_+P) and HPP-treated cheeses stored under vacuum in plastic film (Ch_P_+V) or wrapped in paper without vacuum (Ch_P_+P) at 0, 3, 6, and 10 months of storage.

**Table 1 foods-12-01935-t001:** Moisture (% g 100 g^−1^), protein content (% g 100 g^−1^), pH values, and titratable acidity (g _lactic acid_ 100 g^−1^) measured at 0, 3, 6, and 10 months of refrigerated storage of *Serra da Estrela* control cheeses stored under vacuum in plastic film (Ch_C_+V) or wrapped in paper without vacuum (Ch_C_+P) and HPP-treated cheeses stored under vacuum in plastic film (Ch_P_+V) or wrapped in paper without vacuum (Ch_P_+P).

	Ch_C_+V				Ch_P_+V				Ch_C_+P				Ch_P_+P			
Moisture Content																
0	46.0	±	2.43	^a,A^	46.1	±	0.93	^a,A^	46.4	±	0.39	^a,A^	43.3	±	1.36	^b,A^
3	41.7	±	0.65	^a,B^	41.8	±	1.88	^a,B^	43.3	±	0.39	^a,B^	42.9	±	0.29	^a,A,B^
6	41.7	±	0.73	^b,B^	40.4	±	0.65	^c,B^	43.8	±	0.77	^a,B^	41.8	±	0.56	^b,B^
10	40.8	±	1.47	^a,B^	40.9	±	0.40	^a,B^								
Protein Content																
0	21.9	±	1.83	^a,A^	22.2	±	1.43	^a,A^	24.0	±	0.73	^a,A^	22.1	±	0.47	^a,A^
6	24.9	±	1.20	^a,B^	24.7	±	1.39	^a,A,B^	23.8	±	1.29	^a,A^	24.6	±	1.12	^a,B^
10	24.8	±	0.48	^a,B^	25.3	±	1.30	^a,B^								
pH values																
0	5.24	±	0.01	^a,c,B^	5.19	±	0.01	^b,A^	5.25	±	0.02	^a,C^	5.19	±	0.05	^b,B^
3	5.17	±	0.05	^b,C^	5.18	±	0.01	^b,A,B^	5.82	±	0.05	^a,B^	5.19	±	0.02	^b,B^
6	5.28	±	0.05	^c,B^	5.13	±	0.03	^b,B^	6.69	±	0.09	^a,A^	5.71	±	0.07	^b,A^
10	5.45	±	0.03	^a,A^	5.27	±	0.05	^b,A,B^								
TA																
0	1.43	±	0.10	^a,b,B^	1.39	±	0.04	^a,b,B^	1.30	±	0.02	^b,A^	1.19	±	0.04	^c,B^
3	1.69	±	0.14	^a,A^	1.38	±	0.11	^b,B^	1.20	±	0.07	^c,B^	1.31	±	0.08	^b,c,A^
6	1.58	±	0.06	^a,A,B^	1.25	±	0.11	^c,B^	1.39	±	0.06	^b,A^	1.22	±	0.10	^c,A,B^
10	1.72	±	0.15	^a,A^	1.90	±	0.05	^b,A^								

Values are the mean ± standard error. Different non-capital letters (a, b, c) in the same row indicate statistically significant differences between the same storage time, while different capital letters (A, B, C) in the same column indicate statistically significant differences among the same condition (*p* < 0.05).

**Table 2 foods-12-01935-t002:** Color values of *Serra da Estrela* control cheeses stored under vacuum in plastic film (Ch_C_+V) or wrapped in paper without vacuum (Ch_C_+P) and HPP-treated cheeses stored under vacuum in plastic film (Ch_P_+V) or wrapped in paper without vacuum (Ch_P_+P) at 0, 3, 6, and 10 months of storage.

			Ch_C_+V				Ch_P_+V				Ch_C_+P				Ch_P_+P			
Storage Time (Months)																		
Cheese Surface Colour	** *L** **	0	71.9	±	2.04	^a,C^	73.6	±	2.02	^a,B^	70.6	±	2.24	^a,B^	86.7	±	0.92	^a,B^
		3	77.3	±	1.26	^a,B^	78.7	±	2.04	^a,A^	78.7	±	2.04	^a,A^	77.4	±	1.93	^a,A^
		6	79.8	±	0.35	^a,A^	78.5	±	2.05	^a,A^								
	** *a** **	0	−0.12	±	1.20	^a,A^	−0.43	±	1.31	^a,A^	0.92	±	0.93	^a,A^	−2.93	±	0.14	^a,B^
		3	−0.29	±	1.08	^a,b.A^	0.57	±	1.38	^b,A^	0.57	±	1.38	^a,A^	0.04	±	2.27	^a,b,A^
		6	−0.29	±	0.58	^a,A^	−0.88	±	1.05	^a,A^								
	** *b** **	0	24.4	±	1.20	^c,A^	28.0	±	1.40	^b,A^	25.7	±	1.69	^c,A^	22.3	±	0.74	^a,B^
		3	22.1	±	1.08	^b,B^	24.6	±	1.78	^a,B^	24.6	±	1.78	^a,A^	26.2	±	1.02	^a,A^
		6	20.9	±	0.58	^b,B^	24.3	±	2.05	^a,B^								
Cheese Core Colour	** *L** **	0	85.1	±	2.04	^a,b,A^	85.0	±	1.37	^b,A^	86.3	±	1.83	^a,b,A^	86.7	±	0.92	^a,A^
		3	83.7	±	1.26	^b,A,B^	85.6	±	1.14	^a,A^	85.6	±	1.78	^a,b,A,B^	84.3	±	1.55	^a,b,B^
		6	82.3	±	0.35	^b,B^	82.9	±	1.14	^b,B^	84.4	±	1.32	^a,B^	84.9	±	0.82	^a,B^
	** *a** **	0	−1.54	±	1.20	^a,A^	−1.88	±	0.16	^b,A^	−2.97	±	0.22	^c,A^	0.12	±	1.32	^c,A^
		3	−1.43	±	1.08	^a,A^	−1.34	±	0.10	^a,A^	−3.03	±	0.20	^c,A^	−2.81	±	0.25	^b,A^
		6	−1.49	±	0.58	^a,A^	−1.30	±	0.47	^a,A^	−2.77	±	0.26	^b,A^	−2.47	±	0.10	^b,A^
	** *b** **	0	18.8	±	1.18	^b,B^	22.1	±	1.08	^a,A^	22.6	±	0.98	^a,A^	30.2	±	0.94	^a,B^
		3	19.7	±	1.64	^b,A,B^	20.4	±	1.75	^b,B^	23.5	±	1.08	^a,A^	24.4	±	0.69	^a,A^
		6	21.0	±	1.37	^b,A^	22.2	±	1.44	^b,A^	23.9	±	0.75	^a,A^	24.0	±	0.94	^a,A^

Values are the mean ± standard error. Different non-capital letters (a, b, c) in the same row indicate statistically significant differences between the same storage time, while different capital letters (A, B, C) in the same column indicate statistically significant differences among the same condition (*p* < 0.05).

**Table 3 foods-12-01935-t003:** Textural properties of *Serra da Estrela* control cheeses stored under vacuum in plastic film (Ch_C_+V) or wrapped in paper without vacuum (Ch_C_+P) and HPP-treated cheeses stored under vacuum in plastic film (Ch_P_+V) or wrapped in paper without vacuum (Ch_P_+P) at 0, 3, 6, and 10 months of storage.

Property	Storage Time (Months)	Ch_C_+V				Ch_P_+V				Ch_C_+P				Ch_P_+P			
Hardness	0	0.27	±	0.14	^b,B^	0.18	±	0.04	^b,C^	0.58	±	0.14	^a,A^	0.56	±	0.11	^a,A^
(N)	3	0.47	±	0.093	^a,A^	0.37	±	0.041	^a,b,B^	0.28	±	0.14	^b,B^	0.39	±	0.046	^a,b,B^
	6	0.49	±	0.085	^a,A^	0.59	±	0.13	^a,A^	0.28	±	0.079	^b,B^	0.59	±	0.10	^a,A^
	10	0.30	±	0.047	^a,B^	0.35	±	0.084	^a,B^								
Consistency	0	1.7	±	0.41	^b,B^	1.4	±	0.34	^b,C^	4.8	±	1.36	^a,A^	5.1	±	0.87	^a,A^
(N s^−1^)	3	3.6	±	0.67	^a,A^	2.9	±	0.61	^a,B^	1.3	±	0.47	^b,C^	3.1	±	0.65	^a,B,B^
	6	4.1	±	0.79	^a,A^	5.1	±	0.67	^a,A^	2.8	±	0.91	^b,B^	5.0	±	0.92	^a,A^
	10	2.4	±	0.33	^a,B^	2.7	±	0.33	^a,B^								
Adhesiveness	0	0.6	±	0.19	^a,A^	0.3	±	0.07	^a,A^	1.3	±	0.61	^b,A,B^	1.4	±	0.09	^b,A^
(N s^−1^)	3	1.5	±	0.44	^b,B^	1.1	±	0.41	^a,b,B^	0.7	±	0.12	^a,A^	1.4	±	0.37	^b,A^
	6	2.4	±	0.58	^a,b,C^	2.3	±	0.47	^a,b,C^	1.6	±	0.69	^a,B^	2.8	±	0.61	^b,B^
	10	1.8	±	0.39	^a,B,C^	1.8	±	0.37	^a,C^								
Cohesiveness (dimensionless)	0	5.5	±	2.2	^a,A^	6.6	±	2.0	^a,A^	2.7	±	0.56	^b,B^	2.6	±	0.49	^b,B^
	3	3.2	±	0.65	^a,B^	4.2	±	0.63	^a,C^	4.4	±	1.8	^a,A,B^	3.9	±	0.46	^a,A^
	6	4.2	±	0.72	^a,A,B^	2.3	±	0.68	^b,B^	5.6	±	1.8	^a,A^	2.6	±	0.32	^b,B^
	10	4.7	±	1.7	^a,A,B^	4.2	±	0.71	^a,B^								
Gumminess	0	1.4	±	0.33	^a,b,C^	1.1	±	0.16	^b,B^	1.6	±	0.16	^a,A^	1.4	±	0.28	^a,b,A^
(N)	3	1.5	±	0.052	^a,B,C^	1.5	±	0.082	^a,A^	1.6	±	0.13	^a,A^	1.5	±	0.043	^a,A^
	6	2.0	±	0.054	^a,A^	1.5	±	0.085	^b,c,A^	1.3	±	0.12	^c,B^	1.6	±	0.16	^b,A^
	10	1.7	±	0.042	^a,A,B^	1.6	±	0.074	^b,A^								

Values are the mean ± standard error. Different non-capital letters (a, b, c) in the same row indicate statistically significant differences between the same storage time, while different capital letters (A, B, C) in the same column indicate statistically significant differences among the same condition (*p* < 0.05).

**Table 4 foods-12-01935-t004:** Values of sensory attributes (scale from −10 to 10) of *Serra da Estrela* cheeses control cheeses stored under vacuum in plastic film (Ch_C_+V) or wrapped in paper without vacuum (Ch_C_+P) and HPP-treated cheeses stored under vacuum in plastic film (Ch_P_+V) or wrapped in paper without vacuum (Ch_P_+P) at 0, 3, 6 and 10 months of storage.

Storage Time (Months)	0				3				6				10	
	Ch_P_+V Vs. Ch_C_+V		Ch_P_+P Vs. Ch_C_+P		Ch_P_+V Vs. Ch_C_+V		Ch_P_+P Vs. Ch_C_+P		Ch_P_+V Vs. Ch_C_+V		Ch_P_+P Vs. Ch_C_+P		Ch_P_+V Vs. Ch_C_+V	
Rind Appearance														
Tonality	−0.59		−0.86		2.41	*	4.22	*	1.86	*	−1.55	*	2.37	*
Homogeneity	−1.38		1.61		−4.58	*	−0.38		3.36	*	0.62		−0.86	
Defects	0.38		−2.59	*	1.98		−4.85	*	−2.67	*	0.01		−0.17	
Paste Appearance														
Color	0.23		−1.48	*	2.80	*	−0.71		1.63	*	−0.99	*	2.38	*
Consistency	−1.24		−2.35	*	3.67	*	2.32	*	1.53		1.49	*	−0.12	
Odour														
Lactic	−0.53		−0.92		−1.60		3.52	*	−2.52	*	−2.04	*	0.63	
Acid	−0.01		0.36		−1.63		0.02		−1.50		−0.27		1.16	
Animal	−0.55		0.42		−0.66		−0.46		−0.30		−2.07	*	0.88	
SCFA	−0.26		−0.24		−1.11		−0.42		0.69	*	−0.23		1.15	
Texture														
Consistency					3.87	*	2.60	*	1.00		2.62	*	2.52	*
Friability					3.00	*	0.27		−0.49		0.56		−0.02	
Unctuosity					3.00	*	−2.66	*	−0.86		−3.97	*	−0.03	
Flavor														
Salty					−2.47	*	−0.39		−1.25		−3.45	*	−0.17	
Acid					3.72	*	2.67		2.06	*	3.74	*	1.23	
Bitter					−1.90		1.31		0.13		−0.85		−0.77	
After-taste					3.43	*	0.06		−0.61		0.01		−0.88	

Data expressed as mean (n = 10); * significant difference (*p*-value < 0.05).

**Table 5 foods-12-01935-t005:** Mean values of sensory attributes (scale from 0 to 10) by classification of *Serra da Estrela* control cheeses stored under vacuum in plastic film (Ch_C_+V) or wrapped in paper without vacuum (Ch_C_+P) and HPP-treated cheeses stored under vacuum in plastic film (Ch_P_+V) or wrapped in paper without vacuum (Ch_P_+P) at 0, 3, 6, and 10 months of storage.

Storage Time (Months)	0				3				6			
	Ch_C_+V	Ch_P_+V	Ch_C_+P	Ch_P_+P	Ch_C_+V	Ch_P_+V	Ch_C_+P	Ch_P_+P	Ch_C_+V	Ch_P_+V	Ch_C_+P	Ch_P_+P
Rind Appearance												
Tonality	2.81 ^a,b^	4.36 ^a,b^	4.36 ^a^	2.53 ^b^	1.13 c	5.00 ^b^	7.96 ^a^	3.68 ^b^	0.91 ^d^	2.61 ^c^	9.1 ^a^	6.65 ^b^
Homogeneity	4.94 ^a^	2.26 ^b^	3.01 ^a,b^	4.31 ^a,b^	7.36 ^a^	7.66 ^a^	0.94 ^b^	6.66 ^a^	6.02 ^a^	6.11 ^a^	3.08 ^b^	3.33 ^b^
Defects	1.38 ^b^	4.20 ^a^	3.54 ^a^	1.20 ^b^	0.71 ^b^	0.69 ^b^	6.26 ^a^	1.04 ^b^	3.50 ^c^	1.72 ^d^	5.64 ^b^	8.15 ^a^
Paste Appearance												
Color	4.65 ^a^	2.44 ^b^	3.83 ^a,b^	2.24 ^b^	3.72 ^b^	4.78 ^a,b^	6.34 ^a^	5.10 ^a,b^	3.90 ^a^	4.39 ^a^	5.44 ^a^	5.91 ^a^
Consistency	5.09 ^a^	2.53 ^b^	4.95 ^a^	2.15 ^b^	4.42 ^b^	6.90 ^b^	3.56 ^a^	5.48 ^a,b^	4.38 ^a^	6.07 ^a^	5.30 ^a^	6.66 ^a^
Odour												
Lactic	3.66 ^a^	2.69 ^a^	3.45 ^a^	2.28 ^a^	3.83 ^a^	4.08 ^a^	3.48 ^a^	2.99 ^a^	4.96 ^a^	3.63 ^a^	3.22 ^a^	2.60 ^a^
Acid	3.41 ^a^	2.19 ^a^	3.13 ^a^	2.32 ^a^	3.30 ^a^	3.59 ^a^	2.13 ^a^	2.74 ^a^	4.35 ^a^	3.24 ^a,b^	1.86 ^b^	1.90 ^b^
Animal	2.74 ^a^	2.56 ^a^	2.52 ^a^	2.73 ^a^	3.22 ^a^	2.22 ^a^	3.31 ^a^	2.97 ^a^	1.40 ^a^	0.94 ^a^	2.98 ^a^	2.33 ^a^
SCFA	2.59 ^a^	1.82 ^a^	2.16 ^a^	1.88 ^a^	2.61 ^a^	1.90 ^a^	1.59 ^a^	3.48 ^a^	1.06 ^a^	1.67 ^a^	1.43 ^a^	0.81 ^a^
Texture												
Consistency					2.50 ^b^	5.64 ^a^	1.49 ^b^	4.61 ^a^	2.94 ^b^	3.96 ^a,b^	3.60 ^a,b^	5.19 ^a^
Friability					1.78 ^a,b^	3.60 ^a^	0.19 ^b^	2.04 ^a,b^	1.32 ^a^	1.32 ^a^	1.42 ^a^	1.28 ^a^
Unctuosity					6.66 ^a^	3.52 ^b^	6.31 ^a^	3.82 ^b^	5.77 ^a^	5.3 ^a^	5.31 ^a^	2.16 ^b^
Flavor												
Salty					4.67 ^a^	4.49 ^a^	4.20 ^a^	3.59 ^a^	4.58 ^a^	3.60 ^a^	3.97 ^a^	2.26 ^a^
Acid					4.40 ^a^	2.84 ^a^	2.84 ^a^	2.64 ^a^	5.65 ^a^	3.75 ^a,b^	3.16 ^a,b^	1.40 ^b^
Bitter					3.93 ^a^	2.49 ^a^	3.59 ^a^	2.97 ^a^	3.00 ^a^	2.94 ^a^	3.62 ^a^	2.98 ^a^
After-taste					5.56 ^a^	3.70 ^a^	5.14 ^a^	3.33 ^a^	5.52 ^a^	4.31 ^a^	4.21 ^a^	2.57 ^a^

Different non-capital letters (a, b, c, d) in the same storage time indicate statistically significant differences (Tukey test *p* < 0.05).

## Data Availability

The data presented in this study are available on request from the corresponding author.

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
