# Peer review of "Comparing Different Packaging Conditions on Quality Stability of High-Pressure Treated Serra da Estrela Cheeses during Cold Storageâ€"

_foods, 2023, doi:10.3390/foods12101935_

Round 1
Reviewer 1 Report
To whom it may concern,
The current manuscript entitles "Comparison of packaging methods and materials on quality stability of high pressure treated cheeses during storage" is interesting since it showes the importance of non-thermal processing in the food technology as well as a reliable processing to preserve the food materilas. However, the manuscript needs more improvement especially in the "discussion" section. After revision, it can be considered for the publication. My specific comments are attached in the PDF file.

Author Response
Author’s answers to reviewers’ comments to Manuscript ID foods-2147399
entitled “Comparison of packaging methods and materials on quality stability of high pressure treated cheeses during storage”
Reviewer #1:
General comments:
“To whom it may concern,
The current manuscript entitles "Comparison of packaging methods and materials on quality stability of high pressure treated cheeses during storage" is interesting since it showes the importance of non-thermal processing in the food technology as well as a reliable processing to preserve the food materilas. However, the manuscript needs more improvement especially in the "discussion" section. After revision, it can be considered for the publication. My specific comments are attached in the PDF file.”
The authors are truly grateful for the reviewer comments and all suggestions were carefully considered and implemented.
L1: The manuscript title was revised and changed accordingly.
L12: The abstract was revised. We have added: “Vacuum-packaging system enabled a better control of cheese proteolysis, which revealed to be closer to original control cheese values at the end of the 10-month storage period. Consequently, cheese stored under vacuum film packaging became harder than non-vacuum paper wrapped cheeses at each time point.” The authors summarized the physicochemical, colour, proteolysis and the sensorial results in the last sentence, since from those results, it can be concluded that for longer storage times, cheese packaging in plastic film under vacuum is preferable, while for shorter periods, non-vacuum paper wrapping system is also a viable option.
L16: The ca. is a written abbreviation of circa (=about).
L23: The keywords were revised and changed accordingly.
L62: The manuscript was revised.
L70: “The authors are grateful for the comment and have changed accordingly
L82 and L99: The manuscript title was revised and changed accordingly.
L235: We thank the reviewer for the comments and have tried to update the information as clearly as possible.
L236: The authors are grateful for the comment and have now included further information concerning the comparison in terms of microbial composition in curd. The lack of reports in the literature concerning the microbial composition of milk from this specific geographical zone hampers a deeper discussion.
The “Error! Reference source not found.” the cross-references to figures and tables were corrected along the manuscript.
L489: Different p-values can be reported according to the level of statistical significance determined.
L546: we confirm the conclusion.

Reviewer 2 Report
This work reported the microbial quality, physicochemical properties and sensorial changes of Serra da Estrela cheese treated with and without high pressure processing, and wrapped with paper/plastic film or non-vacuum/vacuum packed over a 10-month storage. Authors have performed a comprehensive study on the subject matter.
Page 5, Line 184: Check out the unit for gumminess.
Page 5, Line 198: Why only one session of sensory analysis was carried out at 10th month? And which test?
Page 5, Line 201: Space out “packedpackaging”.
Page 6, Line 235: Briefly describe quality of the milk and fresh curd microbial quality at the end of Section 3.1.
Page 9, Line 324: Double check the Error statement. Same comment goes to Page 10, Line 355, Page 11, Line 394, Page 12, Line 403, and many more throughout the manuscript.
Page 9, Table 1: Why data were missing for Chc + P and Chp + P store at 10th month.
Page 11, Line 391: Is it possible to state the acceptable/favourable colour attribute of Serra da Estrela cheese.
Page 16, Line 499: Could the authors further elaborate the observation made?
Author Response
Author’s answers to reviewers’ comments to Manuscript ID foods-2147399
entitled “Comparison of packaging methods and materials on quality stability of high pressure treated cheeses during storage”
Reviewer #2:
This work reported the microbial quality, physicochemical properties and sensorial changes of Serra da Estrela cheese treated with and without high pressure processing, and wrapped with paper/plastic film or non-vacuum/vacuum packed over a 10-month storage. Authors have performed a comprehensive study on the subject matter.
The authors are truly grateful for the reviewer comments and all suggestions were carefully considered and implemented.
Page 5, Line 184: Check out the unit for gumminess.
We added the requested information and now it reads (L197: … and gumminess (N) were calculated.”
Page 5, Line 198: Why only one session of sensory analysis was carried out at 10th month? And which test?
The cheeses non-vacuum packed in greaseproof wrapping paper revealed at 6th month characteristics outside the expected for a Serra da Estrela PDO cheese, having the latter gained a very heterogeneous color. Moreover, the panel considered that ChC+P and ChP+P cheeses at 6 months revealed significantly lower (p < 0.05) acid odour and flavour. These results indicated that ChC+P and ChP+P cheeses were clearly outside of the expected characteristics for this type of cheese, and so the evaluation was ended at this point for these cheeses. (L547- 550)
The study continued with cheeses vacuum packaged in polyamide-polyethylene plastic bag.
Thus, at 10th month one sensory evaluation session was carried out, having the panellists performed a paired comparison test among ChP+V and ChC+V to ascertain possible differences caused by the effect of HPP on cheeses vaccum packed packaging in plastic film.
Page 5, Line 201: Space out “packedpackaging”.
The manuscript title was revised and changed accordingly.
Page 6, Line 235: Briefly describe quality of the milk and fresh curd microbial quality at the end of Section 3.1.
The authors thank the reviewer for the suggestion and have now included related information.
Page 9, Line 324: Double check the Error statement. Same comment goes to Page 10, Line 355, Page 11, Line 394, Page 12, Line 403, and many more throughout the manuscript.
The “Error! Reference source not found.” the cross-references to figures and tables were corrected along the manuscript.
Page 9, Table 1: Why data were missing for Chc + P and Chp + P store at 10th month.
Those cheeses were outside of the expected characteristics for this type of cheese, as previously explained.
Page 11, Line 391: Is it possible to state the acceptable/favourable colour attribute of Serra da Estrela cheese.
According to the book of specifications of Serra da Estrela PDO cheese and NP 1922:1985, the rind should be “light straw yellow, uniform”. Cheese non-vacuum packed in greaseproof wrapping paper at 6 months of storage gained a very heterogeneous color, being outside the expected attributes for this type of cheese.
Page 16, Line 499: Could the authors further elaborate the observation made?
ChC+P and ChP+P cheeses at 6 months revealed significantly lower (p < 0.05) acid odour and flavour. These results indicated that ChC+P and ChP+P cheeses were clearly outside of the expected characteristics for this type of cheese, and so the evaluation was ended at this point for these cheeses.

Reviewer 3 Report
The main objective of the article is to compare the microbiological, physicochemical, physical and sensory characteristics of control cheese and treated HHP, conventionally and vacuum packaging, stored during a long period at 4 ºC. The experimental design is excellent, considering all the variables that allow for solving the main objective.
The abstract has a word limitation, but it does not reflect the results of all the experimental work carried out.
The physical characteristics of the film, such as thickness, oxygen permeability and water vapor permeability, should be included.
In the HHP process, more data about the process control should be included, such as the depressurization time and the temperature reached in the cheese due to the application of high pressures.
Is the Miles and Misra technique (1938) the same as the spread plate technique? For this routine method, maybe it is not necessary to include cite. In addition, the DOI does not correspond to the article.
Why was the determination of the Salmonella pathogen not performed?
Why was the study temperature made at 4 ºC? Currently, other temperatures closer to the reality of its storage are evaluated ().
When reference is made to no considerable change or no statistical differences, the p-value is p ≥ 0.05 rather than p > 0.05.
In conclusion, cheese safety has not been demonstrated. No Challenge studies have been performed.
Author Response
Author’s answers to reviewers’ comments to Manuscript ID foods-2147399
entitled “Comparison of packaging methods and materials on quality stability of high pressure treated cheeses during storage”
Reviewer #3:
The main objective of the article is to compare the microbiological, physicochemical, physical and sensory characteristics of control cheese and treated HHP, conventionally and vacuum packaging, stored during a long period at 4 ºC. The experimental design is excellent, considering all the variables that allow for solving the main objective.
The authors are truly grateful for the reviewer comments and all suggestions were carefully considered and implemented.
The abstract has a word limitation, but it does not reflect the results of all the experimental work carried out.
The abstract was revised. We have added: “Vacuum-packaging system enabled a better control of cheese proteolysis, which revealed to be closer to original control cheese values at the end of the 10-month storage period. Consequently, cheese stored under vacuum film packaging became harder than non-vacuum paper wrapped cheeses at each time point.” The authors summarized the physicochemical, colour, proteolysis and the sensorial results in the last sentence, since from those results, it can be concluded that for longer storage times, cheese packaging in plastic film under vacuum is preferable, while for shorter periods, non-vacuum paper wrapping system is also a viable option.
The physical characteristics of the film, such as thickness, oxygen permeability and water vapor permeability, should be included.
The authors questioned the supplier but without success.
In the HHP process, more data about the process control should be included, such as the depressurization time and the temperature reached in the cheese due to the application of high pressures.
Depressurization took less than 2 seconds, information added in the manuscript (L99). Regarding the temperature reached, as it is a commercial HPP equipment, it does not have temperature control during the actual cycle. Instead, it is only possible to control the inlet water temperature, in which, for the present study, varied between 8 and 10 degrees to minimize lipid oxidation. Although, the current literature points towards a temperature increase of 3-5 ºC degrees for each 100 MPa
So, considering the pressure used, the temperature rise may have instantaneously reached approximately 30 ºC degrees.
Is the Miles and Misra technique (1938) the same as the spread plate technique? For this routine method, maybe it is not necessary to include cite. In addition, the DOI does not correspond to the article.
No, It is a different technique. The reviewer is right. The manuscript was revised and now it reads (L646): “Miles, A.A.; Misra, S.S.; Irwin, J.O. The estimation of the bactericidal power of the blood. J. Pathol. Bacteriol. 1938, 38, 732–749, doi:10.1002/path.1700370208.”
Why was the determination of the Salmonella pathogen not performed?
We are aware that Salmonella may enter milk and the dairy environment from many sources. However, previous studies have demonstrated that Salmonella is not found at high numbers in Serra da Estrela ewe’s milk and that HPP together with the ripened cheese characteristics adequately inactivates Salmonella. Based on this rationale its determination was not included.
Why was the study temperature made at 4 ºC? Currently, other temperatures closer to the reality of its storage are evaluated ().
Taking in account the particular cheese characteristics, cheesemaking preserves at 4 ±
2 ºC degrees, which condition was kept to simulate the cheesemaking factory conditions.
When reference is made to no considerable change or no statistical differences, the p-value is p ≥ 0.05 rather than p > 0.05.
The reviewer is right. The manuscript was revised
In conclusion, cheese safety has not been demonstrated. No Challenge studies have been performed.
The objective of this study was to demonstrate the impact of different packaging systems on the maintenance of cheese quality parameters, including microbial evolution. The study enabled to demonstrate that differences are observed and that the selection of the packaging system is dependent on the length of the storage period. Quality parameters were constantly benchmarked with available data in the literature and we were able to show that HPP treatment combined with the proper packaging system contributed to a good standard Serra da Estrela cheese comparable to those commercially available.
